# Effect of Narrative Intervention with Strategy Instruction on the Listening and Reading Comprehension of Children with Autism

**DOI:** 10.3390/bs15081020

**Published:** 2025-07-27

**Authors:** Trina D. Spencer, Megan S. Kirby

**Affiliations:** 1Juniper Gardens Children’s Project, Department of Applied Behavioral Science, University of Kansas, 444 Minnesota Ave. Suite 300, Kansas City, KS 66101, USA; 2Applied Behavior Analysis & Autism Studies, Mary Baldwin University, 101 W Frederick St., Staunton, VA 24401, USA; mekirby@marybaldwin.edu

**Keywords:** oral language, narrative intervention, listening comprehension, reading comprehension, strategy instruction

## Abstract

Some children with autism may require additional support to meet academic expectations for comprehension. Because an extensive set of research links oral narration to listening and reading comprehension, the promotion of narrative-based skills may be a viable intervention approach. The purpose of this study was to examine the effect of narrative intervention with explicit strategy instruction on the listening and reading retells of children with autism after hearing and decoding novel stories. Four children with autism aged 7 and 9 years old participated in this multiple baseline across participants single-case experimental design study. Behavioral therapists delivered the narrative intervention, which included explicit instruction on the use of story grammar icons, to each child individually within the course of their therapy. Results showed that all participants improved their listening (TauU ES range = 0.64–1.06) and reading (TauU ES range = 0.72–1.15) retells, but they required extended use of the icon strategy to achieve the most benefit. When icons were completely removed, three of the four participants performed above baseline levels on the listening and reading comprehension measures.

## 1. Introduction

Narration is a special type of language that, at age five, predicts academic remediation in later primary grades ([38]). The relation between early narrative skills and later reading comprehension is particularly strong ([54]). Children without strong narrative skills are at risk of academic and social problems ([22]). The connection between early narrative skills and reading comprehension exists for children with autism also. For example, [33] ([33]) assessed the narrative and reading comprehension skills of 81 autistic children and adolescents across three time points (15-month intervals). They found that narrative skills at the first time point significantly predicted reading comprehension at the third time point (after 45 months).

Given that narrative skills are clearly linked to reading comprehension, it should cause concern when children do not tell or retell stories in alignment with classroom expectations. Children with autism are among those who may require additional support to understand and produce coherent narratives ([2]; [18]). In research that compared children with autism to typically developing peers or peers with other types of disabilities, children with autism recalled fewer discrete narrative events in a coherent manner ([1]; [12]; [32]), included fewer relevant details ([5]), and recalled less of the story ([72]). Children with autism also produced underdeveloped or incomplete sentences with frequent pragmatic errors when compared to children with typically developing language (e.g., [1]; [32]). In a meta-analysis of 24 narrative studies with children with autism, [4] ([4]) found that peers outperformed children with autism in macrostructure (narrative coherence and cohesive adequacy), microstructure (number of words, number of different words used, and number of utterances), and internal state language (talk about perceptions, thoughts, beliefs, and feelings).

The same discourse and sentence-level structures (i.e., macrostructure, microstructure, and internal states) exist in both oral and written modalities of storytelling, suggesting that narratives can be used in the assessment of both listening and reading comprehension. There is a large body of evidence indicating listening and reading comprehension are closely aligned higher-order processes (e.g., [3]; [10]; [42]) that draw from the same interrelated skills of vocabulary, inferencing, syntax and grammar, knowledge of discourse structures, attention, and working memory ([27]; [50]; [63]; [69]), all of which are leveraged for storytelling ([11]; [17]; [35]). However, only when the passages are linguistically matched for complexity across listening and reading comprehension conditions can the influence of decoding skills be extracted from the equation, revealing the alignment between listening and reading comprehension ([20]; [67]).

In a more recent investigation of this alignment, [46] ([46]) examined the extent to which listening and reading comprehension were equivalent constructs using an educator- and clinician-friendly assessment called CUBED ([45]). As a general outcome measure ([9]) of K-3 literacy (i.e., for universal screening and progress monitoring in schools), the CUBED contains three subtests aligning with the Simple View of Reading ([20]; decoding, listening comprehension, and reading comprehension): (1) Dynamic Decoding Measures (DDM), (2) Narrative Language Measures (NLM) Listening, and (3) NLM Reading. Of critical importance, the narratives used in the NLM Listening subtest were carefully equated with the narratives used in the NLM Reading subtest, meaning they have the same lengths, discourse structures, sentence complexity, vocabulary, etc. Ensuring the complex literate language typical of written text is accessible via spoken language eliminates the potential confound of mismatched linguistic complexity of what is heard and what is read. The researchers administered all three subtests to a diverse group of second- and third-grade students. In the NLM Listening and NLM Reading, students retold stories and answered questions related to the story; the only difference was that in the listening context students listened to the story read to them and in the reading context students decoded the story themselves. They found strong evidence of construct equivalence (*r* = 0.75, *p* < 0.001) and confirmatory factor analysis resulted in listening and reading comprehension loading into a single factor ([46]).

Although parallel stories have not been used to examine listening–reading comprehension alignment in children with autism, there is consistent evidence that both modalities are supported by oral language skills ([7]; [31]; [36]). Importantly, despite often demonstrating age-appropriate or even advanced word reading skills, many children with autism show significant weaknesses in listening and reading comprehension ([6]; [34]; [51]). This dissociation highlights the importance of language-based comprehension interventions for this population. Indeed, a growing body of research has targeted the comprehension difficulties of children with autism, particularly through interventions that support inferencing, vocabulary, and discourse processing ([13]; [53]; [55]).

Because storytelling requires the integration of linguistic, cognitive, and social-pragmatic skills, areas often vulnerable in autism, oral narrative intervention offers a promising format for promoting comprehension among school-aged children with autism. Narrative interventions are specialized language interventions that feature some type of oral storytelling practice as an active ingredient and where the interventionist intentionally promotes children’s language at the word, sentence, or discourse level ([59]). Although storybook interventions may address narrative-related outcomes, there is evidence to suggest oral storytelling produces superior outcomes compared to book reading ([47]; [65]).

For children with autism, narrative intervention may provide a particularly effective mechanism for improving both listening and reading comprehension by supporting the integration of language, memory, and social-cognitive processes ([36]). The structured and repetitive nature of narratives, coupled with explicit instruction in story grammar, can scaffold the development of coherence-building and inferencing skills, which are known to be challenging for children with autism ([12]; [31]). Furthermore, by making the global structure of narrative discourse explicit and visually accessible, narrative intervention helps children with autism organize incoming linguistic information, identify temporal and causal relationships between events, and develop a mental model of story content ([30]). This supports the construction of situation models during listening and reading and is essential for deep comprehension ([73]; [50]).

### 1.1. Narrative Interventions for Children with Autism

There are a handful of narrative intervention studies that feature retelling as a key teaching procedure with autistic children with autism (e.g., [6]; [8]; [14]; [68]). For research outcomes, fictional retelling or personal story generation are common (see [48]). For example, [14] ([14]) studied the preliminary effectiveness of an individualized intervention aimed at improving the fictional narrative retells of four children with autism (6–9 years old). Participants received individualized instruction across 21–52 sessions, with lessons involving visual supports, modeling, and story retell practice. The narrative intervention led to three out of four participants making moderate to large gains in retelling a complete and coherent fictional story with and without visual supports for key story grammar elements (e.g., character and setting). However, the effects did not generalize to retellings of narratives read aloud from classroom storybooks.

One important limitation of research in this area is the repeated use of a single story during intervention, with the accuracy of that story retelling a criterion for introducing a new narrative. As an example of this, [68] ([68]) conducted a multiple baseline across stories design to test the effectiveness of using backward chaining with leap ahead procedures ([61]) to teach children with autism between 4 and 8 years old to orally retell trained stories. During the intervention, each story was divided into segments and the children were asked to retell the last segment of a story. Once a participant met mastery criteria for the last segment, they were required to retell the last two segments. The procedures continued until the participant could retell the complete story independently. Story retelling scores increased for all three participants, although minor procedural modifications were required for two out of three participants. [6] ([6]) replicated the study with three children diagnosed with autism between 8 and 11 years old. Two participants improved their retelling of the trained stories. In another study, [8] ([8]) used modeling and echoic and visual prompts to teach children to retell specific stories. A multiple baseline across stories was conducted with four participants: two with autism and two with language delays (not autistic). During narrative intervention sessions, the interventionist read the story to the participants while pointing to scripts and pictures that portrayed the different parts of the story. If a participant did not respond within the allotted time or responded incorrectly, the interventionist re-presented the script and pictures and prompted the participant to repeat the sentences. All four participants (except for one story for participant 1) learned to retell three complete stories.

While rote teaching may have been necessary for those particular participants, improvements that extend beyond those specific stories directly taught are considered more meaningful or socially valid ([64]). In [68] ([68]), evidence of generalization was observed for only one participant. [8] ([8]) did not assess generalization, but since retelling scores remained at zero during the baselines of the second and third stories, we can conclude generalization did not occur. Without generalization to untrained stories, one cannot assume that generative language or generative comprehension skills were affected. A more impressive accomplishment would be for narrative intervention to enable the retelling of untrained and unfamiliar stories, thus promoting a generalizable skill with academic benefits.

### 1.2. Strategy Instruction and Narrative Intervention

Learning to retell a specific story does not necessarily lead to a generalized repertoire of story retelling or build toward listening and reading comprehension. To achieve generalizable story retells, comprehension strategy instruction may be needed. In strategy instruction, interventionists explicitly teach an intermediate strategy that learners apply when presented with novel instances of the same or similar problems ([24]). A popular strategy for boosting comprehension is to teach the story grammar of narratives. There are many variations in story grammars ([21]), but [62]’s ([62]) is among the most common in clinical research. They outlined eight different story grammar elements including setting (includes characters), initiating event (or problem), internal response, internal plan, attempt, consequence, resolution or reaction, and ending. Story grammar strategy instruction involves teaching these parts explicitly using visuals that help to make the abstract discourse patterns more concrete.

Story Champs ([57]) is a narrative intervention program that merges oral storytelling activities with story grammar strategy instruction. This program capitalizes on oral retelling and story generation supported by a consistent set of icons that represent the story grammar elements embedded into the program’s carefully designed stories (e.g., character, setting, problem feeling, action, ending, and end feeling). In every lesson and across many stories, the same story grammar icons are used to make the abstract and invisible story structure more concrete and visible. Interventionists teach the icons explicitly and pair them with the auditory parts of the stories as well as story-specific illustrations. The goal is to fade the icons systematically to facilitate the transfer from storytelling with visual supports to storytelling without them. This fading procedure, coupled with Story Champs’ 24 different stories, makes it potentially better than other story grammar interventions for building independent and generative comprehension skills. However, maximum generalization is dependent on fading the strategy. Successful transfer of the strategy was observed in Story Champs studies when preschoolers with disabilities applied their knowledge of narrative structure (without the icons) to retell a story they had never heard ([60]) and school-age children with autism generated original personal stories without the story grammar icons available ([44]).

The previous studies showed that Story Champs, with its story grammar strategy, led to improved listening comprehension (i.e., retelling after hearing) and personal story generation. In the current study, we were primarily interested in its effect on reading comprehension (i.e., retelling after decoding) of children with autism. It was not clear, however, if listening comprehension and reading comprehension would be differentially impacted. Therefore, the following research questions were addressed in this study: (1) To what extent does an oral narrative intervention with embedded strategy instruction improve listening comprehension performance of school-age children with autism? (2) To what extent does an oral narrative intervention with embedded strategy instruction improve reading comprehension performance of school-age children with autism?

## 2. Materials and Methods

### 2.1. Research Design

We employed a multiple baseline across participants single-case research design to investigate the effect of Story Champs on listening and reading comprehension. Single-case research methodology relies on within-subject demonstrations of effect that are displayed on graphs. These data are then interpreted through visual analysis of changes in level, trend, and variability. [26] ([26]) outline the minimum standards for single-case research to be considered rigorous evidence. To be regarded as quality evidence, a study must include at least five data points in each condition used to determine causal effect (baseline and treatment). In addition, a study must allow for at least three observations of that change in effect at different times. In a multiple baseline design across participants study, each participant experiences baseline and treatment conditions but the change from those conditions occurs at different times for each participant, shown across the x-axis. In other words, the intervention is applied to each participant at different times in the study, which yields a different number of baseline sessions for each participant, with the smallest number being five needed to establish stability ([28]). The current study meets the standards for quality evidence by including four participants, allowing for four observations of change, and by including at least five data points in baseline and treatment conditions.

### 2.2. Participants and Setting

Participants were recruited from a U.S. clinic that provided services to children with autism based on the science of behavior. Behavior analysts from the clinic sent flyers home with their clients who they believe met the inclusion criteria and would benefit from comprehension-focused intervention. Children could be included if they (1) had an autism diagnosis, (2) scored within level three on the Verbal Behavior Milestones Assessment and Placement (VB-MAPP; [66]), and (3) decoded at least 40 words a minute on the Narrative Language Measures (NLM) Reading ([45]). Children who engaged in behavior(s) that routinely interrupted service delivery and threatened the health and/or safety of themselves or others (e.g., physical aggression), as well as children who lived outside of the U.S., were excluded from the study.

Children of the parents who returned signed consent forms were further evaluated according to the inclusion criteria. Behavior analysts who supervised the children’s cases conducted record reviews to confirm that participants had been diagnosed with autism by a physician or psychologist and that their most recent VB-MAPP scores were within level three (i.e., receptive and expressive language and learning milestones representative of children aged 30–48 months; [39]). Researchers assessed potential participants’ word reading and story retelling skills using the NLM Listening and NLM Reading. Per parent and clinician verbal reports to the research team, none of the participants exhibited severe problem behavior (no physical aggression toward self or others) on a routine basis, but all required behavioral strategies (e.g., token economy systems, regular breaks and/or redirections, first/then activity schedules) during instruction to increase attention and maintain motivation. Each participant’s parent(s) answered demographic questions regarding their ethnicity and language(s) spoken at home.

Martin was a 7-year-old White male who spoke English at home. He received instruction in a public general education classroom, always accompanied by a one-on-one paraprofessional. Martin used 1–3 word-phrased speech to make requests and could answer simple YES/NO questions (e.g., Are you ready? Do you want to stop?). He often engaged in echolalia, repeating the last 2–3 words said aloud by an adult talking to him.

Queen was a 7-year-old White female whose family spoke English at home. She attended a private special education day school during the week and received outpatient applied behavior analysis therapy services during her school day. Queen used 3–4 word sentences to make requests, answer basic questions, and ask questions (e.g., Can I be done?). She had some articulation problems that could make it difficult for unfamiliar adults to understand her.

Mario was a 9-year-old, Middle Eastern, White male who received most of his first-grade public education in a general education classroom, with the exception of related services, which were delivered in a specialized setting. His family spoke Arabic at home, but Mario primarily used English to answer questions with short phrases (2–3 words). Mario often sang to himself and repeated questions or directives asked of him by adults in lieu of answering WH-questions about stories he read aloud.

Serena was a 7-year-old African American female who attended a public school and received instruction within a general education first-grade classroom. Her family spoke Swahili at home, but she preferred to speak in English. Serena was able to use complete sentences with some stuttering, but most speech was scripted and focused on a topic of her interest. For example, when asked to retell a brief single-episode story read aloud to her, Serena usually provided the basic plot to an unrelated story from a favorite TV show.

Research activities (data collection and intervention sessions) took place during children’s regularly scheduled individual services in a clinic room. The room was shared with other children who also received services but were not involved in the study. All procedures were conducted on a tabletop while the interventionist and child sat across from each other or side-by-side. Serena participated in some sessions in her home because she received part of her services in that setting. The intervention procedures were conducted in her room on the floor, which was her preferred location for therapy.

### 2.3. Research Team

The research team consisted of interventionists, research assistants (RAs), an independent data analyst, and a project manager. The interventionists were three registered behavior technicians (RBTs) and one master’s level Board Certified Assistant Behavior Analyst (BCaBA) who worked at the clinic. They collected the story retellings and delivered the interventions with their clients. RAs were undergraduate students studying applied behavior analysis or communication science and disorders. They scored participants’ retells from recorded audio files. The independent analyst reviewed the graphs and made data-based decisions regarding condition and phase changes. Decisions were based on the number of sessions from when the previous participant entered intervention and the stability of data patterns.

Prior to any research activities, the project manager trained the interventionists on the assessment and intervention procedures. She modeled the intervention and role-played with the interventionists until they reached 100% fidelity. Additionally, she observed their collection of listening and reading retells and the delivery of the intervention with other children in the clinic who were not research participants. RAs were required to score practice retells until they demonstrated 90% agreement with a master scorer on three retells.

During the treatment condition, the project manager observed 30% of the interventionists’ sessions to measure intervention fidelity using the same 16-step procedural checklist used for training. When observing the delivery of intervention sessions, they placed a checkmark next to each step implemented correctly. To calculate fidelity, the total number of steps completed correctly out of 16 was converted to percent. The mean fidelity score for all interventionists was 98% (range: 96–100%). At the participant level, mean intervention fidelity across sessions was as follows: 96% for Martin (range: 95–100%), 96% for Queen (range: 94–100%), 100% for Mario, and 100% for Serena.

### 2.4. Intervention Materials

The Story Champs ([57]) program features 24 carefully constructed stories so that every story contains the planned story grammar elements, complex sentences, and a child-relevant theme (e.g., wanting a toy someone else has, getting hurt, food preferences). There are 10 levels of stories (A–J). Given the age of the participants (7–9-year-olds), we selected the level that would be appropriate for 6–7-year-olds—Level B. Every Level B Story Champs story contains roughly the same number of words (125–135) and the same story grammar elements of character, setting, problem, feeling, plan, action, consequence, ending, and end feeling. Stories have Lexile ratings between 500 and 510 and a mean length of utterance of 9.2. Each story was accompanied by a set of five simply drawn illustrations with minimal color elements that reflected the content in the story: (1) character and setting, (2) problem, (3) feeling, (4) plan, main character’s action (attempt) and what the secondary character did to help (i.e., consequence), and (5) ending and the end feeling.

At Level B Story Champs, only seven icons are taught to children (i.e., character, setting, problem, feeling, action, ending, and end feeling), although nine story elements are written into the stories. Each of the icons has a unique color and simple graphic to represent each story grammar element (samples can be viewed at www.LanguageDynamicsGroup.com). Additional materials include a one-page lesson plan containing directions to the interventionist and the steps to deliver the intervention (see below).

### 2.5. Dependent Variables and Measurement

The dependent variables in this study were listening and reading comprehension, measured through retells. Stories from the NLM Listening and NLM Reading subtests of the CUBED assessment suite (downloadable for free at www.LanguageDynamicsGroup.com) were used to elicit retell language samples from participants ([45]). Although the NLM Listening and NLM Reading contain several parallel stories per grade, only the first-grade stories (appropriate for children 6–7 years old) were used to measure comprehension via narrative retells in the current study. The first-grade NLM Listening includes 25 stories and the first-grade NLM Reading includes a different set of 25 stories. However, all the stories at first grade are structured the same to ensure valid and repeated sampling of listening and reading comprehension. The stories contain 130–135 words and the exact structures described for the Story Champs Level B stories above, with a Lexile rating between 500 and 510 and a mean length of utterance of 8.81 ([45]). The NLM stories are considered novel and unfamiliar because they are distinctly different from the 24 Story Champs stories used for intervention, and NLM stories were never used during intervention sessions. Both assessments have sound validity (r = 0.63–0.88) and reliability (82–100% scoring agreement) ([45]; [46]).

#### 2.5.1. Administration of the NLM Listening and NLM Reading

To elicit information about a child’s listening comprehension, the interventionist read a story to the child while the child listened to the story without any visuals. The interventionist said, “Thanks for listening. Now you tell me that story.” During the child’s retell, the interventionist used standardized neutral prompts to encourage the child, such as, “It’s OK. Just do your best,” and “I can’t help, but you can just tell me the parts you remember.” The elicitation of one listening retell took 2–5 min.

The NLM Reading was conducted in the same manner, with the exception that the child read the story before retelling it. Scripted administration included the following, “Please read this out loud. Do your very best reading. I’ll help you if you need it. When you’re done, I might ask you to tell me the story.” Once the child read the story, interventionists asked the child to retell the story, “Thanks for reading. Now you tell me that story.” Standardized prompts were used during the reading retells. The elicitation of one reading retell took 5–8 min.

#### 2.5.2. Retell Scoring

RAs scored children’s story retells while listening to the audio files. NLM Listening and NLM Reading record forms—one page per story—contain story-specific scoring rubrics, with explicit information about what constitutes scores for each story grammar element based on their inclusion and clarity. For example, if a participant included a complete and clear problem in their retell, that would be awarded 2 points. An incomplete or unclear problem would earn 1 point, whereas a zero was given if the story grammar element was missing from the participant’s retell. RAs awarded up to 2 points for eight story elements (i.e., character, setting, problem, emotion, attempt, consequence, ending, and end feeling) and 1 point for the plan, which involved the use of a mental state verb such as decided, considered, imagined, and figured. Although during intervention, only seven story elements were explicitly taught, the assessment stories included nine story grammar elements that could be scored, yielding a total of 17 points possible per retell. It is important to note that because the autistic participants in this study had limited verbal abilities, and the intervention focus was on the macro-discourse elements (i.e., story grammar), their retells were not scored for complex sentences and less common words as is possible with the NLM Listening and Reading tools.

From each condition, we randomly selected 30–33% of each participant’s retells to be scored by an independent RA. For each double-scored retell, we calculated point-by-point agreement by dividing the number of agreements by the total number of agreements plus disagreements, multiplied by 100. Mean percentages were calculated for each measure.

The mean point-by-point agreement of NLM Listening retell scores for Martin was 100% in the baseline condition and 96% in the treatment condition (range: 92–100%). Queen’s mean scoring agreement was 96% and 98% for baseline and treatment conditions (range: 95–100%). Mario’s listening retell scoring reliability results averaged 98% in baseline and 99% in the treatment conditions (range: 97–100%), and Serena’s baseline assessments had a mean reliability score of 100%, and assessments in the treatment condition had a mean of 99% (range: 97–100%).

The mean NLM Reading retell point-by-point agreement for Martin was 95.4% at baseline and 97% during the treatment condition (overall range: 93–100%). For Queen, the scoring reliability of reading retells was 96% and 99% for baseline and treatment conditions (range: 96–100%), respectively. For Mario, the mean scoring reliability was 94% and 99% for baseline and treatment, respectively (range: 93–100%). Reading retell mean reliability scores for Serena’s baseline and treatment were 92% and 99%, respectively (range: 89–100%).

### 2.6. Experimental Arrangement and General Procedures

All participants experienced baseline and treatment conditions. According to concurrent multiple baseline conventions, participants started baseline at approximately the same time (within a few days of each other), and the independent analyst made sure that they started intervention staggered by at least three sessions; however, due to repeated absences of some children and interventionists, individual timelines did not always align; hence, we use sessions rather than dates on the x-axis (see Figure 1 and Figure 2). It should be noted that the Figures’ x-axes do not align with the actual distribution of sessions across time.

For all students, a modification was made to the data collection procedures at the same time. The independent analyst recommended it for all the participants due to a reduction in responding or minimal gains. Because the NLM Listening and NLM Reading capture generalized comprehension skills and are high-level, far-transfer tasks, we anticipated needing a strategy to bridge the intervention and assessment tasks. Therefore, we allowed the children to use the strategy that was taught in the Story Champs intervention (i.e., content-generic story grammar icons) to support the listening/reading and retelling of the stories during data collection sessions. This involved placing the story grammar icons on the table in front of the child before presenting the NLM Listening or NLM Reading story and leaving the icons visible as they retold the story. At the end of the treatment condition, using a novel story, we conducted a post-probe without the story grammar icons available to determine the extent to which generalization to the far-transfer context occurred. Reinforcement systems were used during the assessments and intervention but only for on-task behaviors and engagement with the research tasks, not specific responses.

The masking of the research team increased the objectivity of the data collection, scoring, and analysis. After RAs, who were blind to conditions, listened to the children’s retells and completed the scoring, data were returned to the project manager. She created graphs for the analyst to review and make decisions regarding stability.

#### 2.6.1. Baseline

Two baseline sessions were conducted each week. The NLM Reading and NLM Listening assessments were administered in every session, with the listening assessments preceding the reading assessments. Interventionists provided a 3–5 min break between assessments.

#### 2.6.2. Intervention

Once the intervention condition began for a participant, they received two intervention sessions per week (barring absences). Delivery of the intervention followed the assessment of participants’ listening and reading comprehension within the same session. The intervention procedures lasted 10–15 min, with longer sessions related to participants’ individualized needs and the occurrence of problem behavior.

Four key steps to delivering the intervention were repeated in every session. In the first step, with five illustrations on the table in front of the child, the interventionist read the model story and placed the story grammar icons near the corresponding illustrations. The interventionist named each story grammar element while pointing to the icons and had the child repeat the names of the icons. No other explicit instruction was provided about how to use the icons. Their repeated proximity to the illustrations and contiguous presentation with the spoken story parts were sufficient to teach the icons and their order. In the second step, the interventionist asked the child to retell the story with the illustrations and icons on the table serving as visual prompts. In the third step, the illustrations were removed, and the child retold the story with only the icons available. The icons did not contain story-specific information, only information about the story grammar. In the final step, the icons were removed, and the child retold the same story without illustrations or icons.

At any point during these steps, a standardized two-step prompt was used if the child did not provide a response within three seconds of being asked to retell the story or tell the next part of the story. This involved the interventionist first asking an open-ended question (e.g., “What was his problem?” or “How did she feel about the problem?”). If that was effective, the child was allowed to continue retelling the story. However, if the child was unable to answer the question, the interventionist provided a model sentence for the child to repeat (e.g., “John crashed on his bike. You say that.”). Praise was provided for correct and prompted responses in all steps.

#### 2.6.3. Post-Intervention Probes

When the intervention phase was complete, post-intervention probes were conducted to assess children’s story retelling abilities without the story grammar icons visible in the listening and reading comprehension assessments. These sessions were conducted in the same format as the baseline sessions and allowed for an examination of the bridge strategy’s success.

### 2.7. Data Analysis

We interpreted the results of the study through visual inspection of participants’ graphically displayed data by examining changes in level, trend, and variability from baseline to treatment conditions. In addition to visual analysis, we also indexed the size of the treatment effect using a supplementary statistic, TauU ([41]). TauU is a non-overlap measure commonly used in single-case experimental design studies and meta-analyses ([15]; [40]) that can account for trends in the data while making between-phase pairwise comparisons of baseline and treatment observations. We calculated TauU using the SingleCaseES web application v.0.7.3 ([49]).

## 3. Results

Total scores on the NLM Listening and NLM Reading retells are displayed in Figure 1 and Figure 2, respectively. Two sets of graphs are organized with four staggered baseline lengths across four participants.

### 3.1. Listening Comprehension

Prior to intervention, all participants had low levels of NLM Listening retell scores, as they were unable to retell stories in a coherent and cohesive manner. As seen in Figure 1, following the introduction of the Story Champs intervention, there was generally an immediate change in both data level and trend, but increases were moderate. When participants used the story grammar icons during the assessment sessions, there was a medium-to-large increase in average retell scores, indicating a meaningful improvement in the completeness of their story retells. However, when the post-probe was conducted without icons in the assessment (i.e., far-transfer context), only Serena maintained her gains completely, Martin and Mario partially maintained their gains, and Queen’s retell score returned to baseline levels. We describe each participant’s listening comprehension performance across baseline and treatment conditions below.

#### 3.1.1. Martin

Martin was unable to produce a story retelling during baseline assessments. However, upon introduction of the narrative intervention, there was a small positive change in level as Martin improved his ability to accurately include key story elements during novel story retells when icons were unavailable. As intervention progressed, there was a range of responding and an ascending trend. The TauU effect size (ES) comparing no-icon assessment performance in baseline and treatment conditions was 0.83. After icons were added to the assessment beginning in session 36, there was a large change in level as Martin’s NLM Listening Retell score increased from an average score of 2.07 to an average of 9.88. Comparing treatment outcomes that included the icons during assessment to baseline, the TauU ES was 1.00.

#### 3.1.2. Queen

On 10 out of 11 baseline observations, after listening to a single-episode narrative, Queen was unable to recall any of the story and only included partial information about one story element once (session 4). After intervention began, across four consecutive assessment observations, there was a slight increase in her ability to retell 1–2 parts of the story. However, starting at session 18 and continuing until icons were introduced to the assessment in session 30, Queen did not produce a story retell. Comparing the baseline to the initial treatment observations, the TauU ES was 0.16. Once icons were accessible to her during the assessment, there was a distinct change in the level and variability of Queen’s retelling scores. Her average NLM Listening Retell score increased to 2.46 (range: 0–6) but trended down across the last four observations. Comparing observations in the baseline to the second treatment phase (with icons), the TauU ES was 0.64.

#### 3.1.3. Mario

Mario’s baseline NLM Listening Retell scores were variable for the initial five sessions (range: 0–5), as he sometimes identified the main character, recalled general setting information, and/or the problem. However, his story retelling lacked a complete episode; he did not include information about a character’s actions or attempts to solve their problem (e.g., middle) nor resolution (i.e., ending). Mario could not retell any story details for the remaining seven baseline observations. Although, when intervention started, his ability to retell story information was similar to the level observed at the start of baseline (2.4), having an average Retell score of 2.2. The last three observations of the treatment condition without icons (sessions 20–22) indicated an upward trend, increasing from a score of 2 to 6, as Mario began to include additional story details about character actions and/or endings. The TauU ES comparing the initial treatment phase to baseline observations was 0.61. When icons were available, Mario’s NLM Listening Retell scores were at their highest (average score of 4.77) and he was able to provide at least some episodic details. However, consistent with other phases, his response pattern was variable (range: 0–8) with a TauU ES of 0.94.

#### 3.1.4. Serena

Serena’s NLM Listening Retell scores were at or near zero for most of the baseline condition. On 3 out of 18 baseline observations, Serena retold information about the main character using their name or identifying them using a general noun (e.g., the girl). She provided setting activity and location details once. Once intervention began, her scores increased from a baseline level of 0.28 to an average NLM Listening Retell score of 2.2, and, with the exception of session 23, she consistently provided at least some information about the story. Without icons during assessments, there was a moderate-large treatment effect (TauU ES = 0.79). Once icons were made available to Serena at session 24, there was a positive change in level, and her average NLM Listening Retell score increased to 5.33. In addition to the character and setting, she demonstrated the ability to retell at least some information about the story problem, action, and/or ending, with scores that ranged between 2 and 8. Only one observation in the assessment-with-icons treatment phase overlapped with those in baseline, and the treatment effect is very large (TauU ES = 1.06).

### 3.2. Reading Comprehension

For three out of the four participants, NLM Reading Retell scores occurred at near-zero levels during baseline, indicating poor comprehension of stories read. However, across both the no-icon and icons-in-assessment treatment phases, there was an increase in level for all participants. Further, as shown in Figure 2, most participants immediately or gradually improved upon their retell scores when the story grammar icon strategy was available in the assessments. NLM Reading Retell post-probe data (collected without use of the icons) indicate complete maintenance of treatment gains for Queen and Mario. Martin partially maintained his treatment gains at post-probe. Nonetheless, his post-probe score was substantially above his baseline levels. Only Serena’s reading comprehension performance returned to baseline levels once the icons were removed from the assessment. Participant-specific reading comprehension outcomes are described in greater detail below.

#### 3.2.1. Martin

Across all baseline sessions, Martin was able to independently read a novel single-episode story aloud but could not provide any information about the story during a retell opportunity. After one intervention lesson, there was an increase in level and an upward trend in his ability to accurately and independently retell up to four parts of a story that he read. Between sessions 16–28, however, the positive trend was replaced with significant variability as Martin’s inclusion of key story events ranged from 0 (baseline level) to 2 elements (scores of 2–3). Thereafter, his overall level of retell performance increased and stabilized for several sessions but trended upward with the added access to story grammar icons at session 37. Approximately 23.5% of the baseline and no-icon treatment observations overlapped, resulting in a TauU ES of 0.77, indicating a medium effect. When icons were present during the assessment, there was no overlap between treatment and baseline observations, and the treatment effect was large (TauU ES = 1.00).

#### 3.2.2. Queen

At the time of screening, Queen could read a single episode of a first-grade story with minimal support for decoding. However, when asked to retell it, Queen could not provide any story details or answer questions about the story accurately. Queen’s NLM Reading Retell baseline data were stable, at or near zero. She did not provide a retell after reading a first-grade story with the exception of baseline session 5. With the introduction of the narrative intervention, Queen’s ability to read and retell a story without use of icons immediately increased in level from 0.09 to 1.94, and to 3.18 when icons were used in assessment. Across the treatment condition, however, there was significant variability and Queen failed to retell any story details on nine occasions. The overlap between baseline and initial treatment observations was moderate (50%), with a medium effect size for no-icon assessment conditions (TauU ES = 0.63). Conversely, when icons were used in the assessment, the degree of overlap was smaller (TauU ES = 0.72).

#### 3.2.3. Mario

During baseline, when asked to read and retell a first-grade story, Mario was able to produce details about 1–2 story elements on four occasions; he provided general information about the problem, character, and/or action once, and included a clear and complete ending twice. Baseline scores ranged from 0 to 3. After the intervention started, Mario independently and consistently produced information about two or more story parts. His average score increased from a baseline level of 0.83 to 3.80, although there was slight variability in the completeness of retold information (range: 2–7), with a medium-sized treatment effect (TauU ES = 0.82). Once Mario had access to icons in assessments, there was an immediate increase in level and an upward trend in scores. His average NLM Reading Retell score increased to 8.54, and he routinely demonstrated the ability to retell a story with a complete episode (i.e., problem, action, consequence and/or ending), with only one pairwise data overlap between baseline and assessment-with-icons observations. The estimated treatment effect was large (TauU ES = 0.97).

#### 3.2.4. Serena

Similarly to her performance on NLM Listening baseline assessments, Serena was only able to provide accurate information about a story she read on two baseline opportunities, and details were restricted to the main character. Her average baseline level was 0.22, but when intervention began, Serena’s ability to independently read and retell her scores increased to an average of 4.2. While she did not retell any details about stories read on two occasions, she independently provided clear and complete information about the story character, setting, problem, and/or ending in sessions 20–22. Although there was partial data overlap, the degree of improvement across the first five treatment observations was associated with a medium-large treatment effect (TauU ES = 0.82). After icons were introduced to the assessment condition at session 24, her retell observations increased to an average level of 6.44. Although there was a moderate degree of variability, there was no overlap between the baseline and assessment-with-icons treatment scores, and the estimated treatment effect was very large (TauU ES = 1.15).

## 4. Discussion

The purpose of this study was to examine the effect of narrative intervention with story grammar strategy instruction on the listening (research question 1) and reading (research question 2) retells of children with autism. As a common activity in school, retelling reflects the ability to understand and to express one’s understanding. For many children with autism, this can be particularly challenging because retelling demands high-level oral language skills ([29]).

The results indicate that narrative intervention led to small gains in listening and reading comprehension without the use of the icons during assessments, but when participants were allowed to use the comprehension strategy we taught them, their gains were noticeably larger. This study was not designed to isolate the effect of the story grammar icons from the other teaching procedures. Nonetheless, the post-probe data help answer the research questions. In terms of research question 1, TauU estimates of effects indicated moderate to large improvements in listening comprehension (range = 0.64–1.06) related to the oral narrative intervention, and when the icons were unavailable during the post-probe retell, Queen was the only participant who did not show at least partial maintenance of gains. For research question 2 about the effect of narrative intervention on reading comprehension, TauU estimates of effects indicate moderate to large gains (range = 0.72–1.15), and Serena was the only participant who did not maintain any of her gains. These results are encouraging and reflect moderate to strong effects in general, indicating that autistic children can acquire generalized comprehension skills.

As autism affects individuals differently, it is not surprising that the results are not uniform across these four participants. Because we did not gather detailed learner profiles and we only had four participants, it is difficult to pinpoint individual characteristics that may account for Queen’s difficulty with listening comprehension without the icons and Serena’s difficulty with reading comprehension without them. It was also unfortunate that we were unable to extend the final condition without the icons. Due to a significant holiday break, we were only able to collect one post-probe listening retell and one reading retell from each participant. Given that both Queen and Serena demonstrated variable responding throughout the study, it is impossible to know whether their zero post-probe scores were evidence of an inability to retell without the icons present or if there were idiosyncratic influences at play.

At a high-level interpretation, we can say that participants demonstrated reasonably similar baseline and treatment patterns of responding in the listening and reading comprehension contexts. In other words, their retell abilities after reading generally corresponded to their retell abilities after listening. There are a few possible exceptions to this, however. Although Marios’ baseline listening and reading retells were comparable, the narrative intervention yielded consistently larger gains for him in reading comprehension (mean of 6.48) than in listening comprehension (mean of 3.65). Queen and Serena also exhibited slightly better results for reading comprehension than listening, although average differences of 1.28 and 1.42, respectively, may not be meaningful.

Several researchers have theorized why comprehension of written material might be stronger than comprehension of spoken language. Written text remains visually available, allowing readers to pause, reread, and control the pace of input, whereas spoken language must be processed in real time, placing greater demands on working memory ([43]; [73]). For children with autism, these differences may be particularly salient. Some may benefit from the predictability and visual stability of print ([37]; [56]). Thus, small advantages in reading comprehension over listening comprehension, as observed with Mario, may reflect modality-specific processing differences that are especially relevant for populations with known difficulties in auditory attention or inferencing from spoken input.

### 4.1. Meaningful Findings

One of the most important findings was that greater retell scores were observed when the visual supports were available to the children than when they had no visual supports. This was not surprising because similar researchers concluded that many children with autism will likely require visual supports during comprehension activities ([25]; [71]). With respect to our findings, there are a few interpretations worth mentioning. First, the Story Champs intervention and the minimal explicit instruction on the use of the icons were sufficient to teach children how to use the icons when presented with novel stories. The icons did not provide children with information about specific stories, yet improved their retells of novel stories. It is possible that the participants, and maybe other children with autism, have difficulty organizing narrative content but may not have difficulty with the content itself. In a scenario like this, strategy instruction may be most fruitful. We know that strategy use can be extremely powerful, especially for students who require accommodations to be successful in general education settings ([52]). A strategy like the use of icons can be available to children on a notecard, affixed to a whiteboard, or in a worksheet. Children who have been trained to use the icons can access them when necessary, resulting in a highly feasible academic modification. During the first intervention phase (without the icons in the assessments), several participants asked for the icons or said, “Where are the colors?” or “I need the circles.” This suggests the children in our study recognized that they needed more support and were comfortable using the icons in this manner.

A second meaningful finding is the extent of transfer achieved. In some of the previous intervention studies with autistic children in which retelling was the primary dependent variable, none of the participants generalized their retelling skills to novel, untaught stories ([6]; [8]; [68]). In the current study, transfer from trained stories to untrained and unfamiliar stories occurred somewhat rapidly, although icons were needed to continue children’s retelling improvements. Another far-transfer achievement was observed in the reading retell data. The generalization from listening retells, as practiced during intervention, to reading comprehension outcomes is noteworthy. Despite observing modest transfer from the use of icons to retelling in their absence, an oral language intervention that can improve reading comprehension of children with autism has immediate and important implications for practice. Improving both sets of repertoires while only explicitly teaching one suggests oral narrative intervention is efficient and powerful. This outcome also extends the findings of [58] ([58]), who found the oral narrative intervention impacted children’s writing, including one child with autism, and contributes to the larger body of research on reading comprehension interventions for this unique population ([13]; [16]; [33]).

To affect change in generative comprehension skills, it was essential that children do not practice the same story over and over (e.g., [6]; [8]; [68]). The use of many different stories was an intentional design feature of Story Champs ([57]) so that generalization to untrained stories and contexts could be achieved. In this study, children practiced different stories during intervention sessions and never heard the same story in consecutive sessions or even consecutive weeks. This type of multiple exemplar practice likely led to children’s ability to retell untrained stories. The gradual fading of visual supports, including the illustrations and the icons, was another intentional feature designed to promote generalization. It is important to note that the illustrations portrayed moderate information about the story because they depicted specific story scenes, whereas the story grammar icons portray no information about a specific story. Icons merely depicted the organization of stories generally; thus, they were generic visual prompts.

Despite the important contribution this study makes regarding strategy use, the ultimate goal is to prepare children with autism for complex academic activities. We consider retelling in the absence of all types of prompts to be the best possible outcome. It may be necessary to explore a handful of possible modifications to achieve a complete and far transfer to retelling without icons. One possibility would be to monitor children’s progress during intervention sessions as other researchers had performed ([6]; [8]; [68]). Our assessment data represented intermediate maintenance (>2 days) and generalization (novel stories), not just acquisition. An approach that is more sensitive to subtle intervention effects would be to collect retell data during the final step of the intervention in which the illustrations and icons have been removed; however, such an assessment would involve trained stories, reducing its social significance. An alternative is to conduct a probe with a novel story immediately following the intervention session (e.g., [44]). This may facilitate a more gradual transfer toward retelling a novel story days later. Finally, a different type of strategy could be used during the intervention, such as gestures that correspond to the story grammar elements. The Story Champs large and small group procedures include a gestures game that could be used in the individual intervention too. Gestures may be an appropriate alternative strategy because the children would have control of their use and no additional materials would be needed ([70]).

### 4.2. Implications for Practice

A vast literature describing the feasibility and efficacy of Story Champs exists (see Research Synthesis at https://www.languagedynamicsgroup.com/story-champs-2/story-champs-research/ (accessed on 22 May 2022)), and the results of this study can be considered in the context of the larger body of evidence. The Story Champs program is regularly used in the U.S., Australia and Canada in general education classrooms and by special educators to support the language-based literacy skills of diverse children. The carefully constructed stories with multiple difficulty levels, strategic visual materials, and explicit teaching procedures make the program easy to implement and versatile. While Story Champs has been used effectively with a range of children with learning and developmental disabilities ([19]; [60]), including children with autism ([44]), the children in the current study represent a group with higher support needs than those in previous research. Therefore, there are a few important implications to consider for practice.

Clinicians need not be hesitant about using a sophisticated form of language intervention, like narrative intervention, with children with high support needs. All the children in the current study received intensive behavioral services regularly and had individual behavior plans in place. It may, however, require more sessions to reach a meaningful magnitude of effect and gains may not be as large as expected for children with low support needs, as evidenced in this study. It is likely that the higher the support needs, the more intervention sessions needed. While on its own this is not problematic because many children with autism receive intensive, individual therapy, there are a few points of caution for clinicians. First, if the children are not successful when prompts are faded, they may stop responding. We saw this occur with Queen and Serena during the intervention phase. For them, the intervention faded the prompts too quickly and they became frustrated by the lack of visual support. We used the icons to help reduce the jump from the supportive intervention to the unsupported assessment context, and immediate improvements in retells occurred, as shown in Figure 1.

The second caution is related to the repetition of stories. Story Champs is designed to have many exemplars and for many children, the structures can be taught before exhausting all 24 stories. For children with high support needs, however, the stories may eventually need to be repeated. However, it is very important that stories are not repeated on consecutive sessions but recycled once the entire 24 stories have been used. It is tempting to give children with high support needs concentrated repeated practice, but that is counterproductive because that encourages memorizing the content. The goal of Story Champs is to produce generative language abilities, and in the case of the current study listening and reading comprehension of novel stories was achieved.

Another implication for practice is that the story grammar icon strategy may be useful in many contexts and fade on its own. While autism practitioners should strive to withdraw all types of prompts, this one might not be too problematic when it is not completely faded. Children can potentially use the icons as a just-in-time support when they need it, how they need it, and for as long as they need it. Individuals with high support needs often benefit from strategies ([24]) to support their successful engagement in other tasks. Once the story grammar icons have been taught, displaying them on a table or a wall in view could be sufficient. Children with autism might be able to take a quick peek during literacy tasks and rely on them momentarily to help them organize narrative content for production or comprehension tasks.

### 4.3. Limitations and Future Directions

The results of this study contribute to several areas of research, including narrative intervention, strategy instruction, and comprehension. However, it is important to note there are some limitations that impact what conclusions can be drawn. The intervention has many components with two primary active ingredients—retelling and story grammar icons. Story Champs capitalizes on retelling practice during the intervention, which is supported by icons. Because the intervention is conceptualized as a multiple component package, the icons were not presented during baseline. It is not standard practice to include components of the intervention package in baseline assessments (e.g., [8]; [68]). However, the necessity of the use of icons in the later phase of this study suggests there is a slight possibility the icons could have had a similar effect without the retelling practice. From the current study, we cannot confirm what children’s retell performance would have been in the baseline condition had the icons been available. While it is possible that children could have had higher baseline retell performances if they had been allowed to see the icons, it is not very plausible. The icons are neutral, abstract, and symbolic stimuli that require teaching and association with stories to learn their meaning. Without additional teaching or practice using the icons, it is unlikely that participants would be able to use them effectively. Given the theory of change for Story Champs, and other narrative interventions, the most potent teaching procedure is most likely the multiple exemplar story retell practice. It is believed that the icons only become useful (i.e., conditioned) through their association with several stories. They are included in Story Champs to facilitate the gradual removal of visual supports. Evidence of children using the icons as a just-in-time strategy to bridge between storytelling with pictures and storytelling without pictures was seen in [23] ([23]) when the children drew the icons on their papers without being directed to or taught to do so.

Although we cannot ascertain the effect of icon use without teaching or practice, the combination of retelling practice and the use of the icons contributed to improvements when the icons were not available. Nonetheless, improvements were more pronounced when they were available in the assessments. The purpose of the study was not to examine the components of the intervention, but that is likely what is needed next. Future research should examine the individual ingredients of narrative interventions to determine their utility and necessity for realizing key academic outcomes. At this point, however, a reasonable conclusion is that the use of icons is a useful comprehension strategy, and practice retelling with the support of the icons may enhance children’s ability to use the icons in a generalized fashion.

## Figures and Tables

**Figure 1 behavsci-15-01020-f001:**
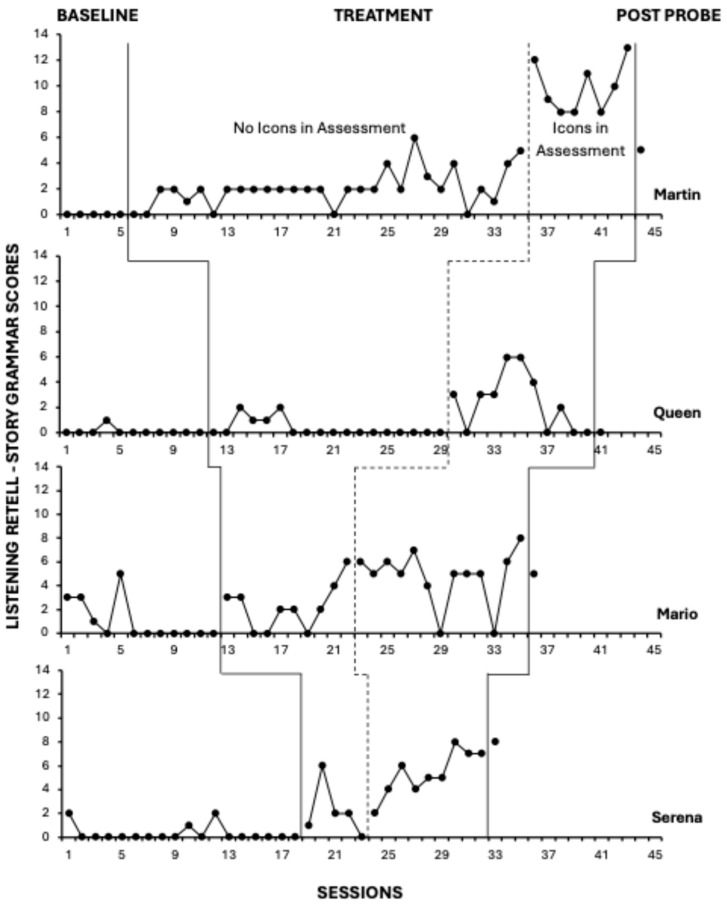
Listening comprehension results.

**Figure 2 behavsci-15-01020-f002:**
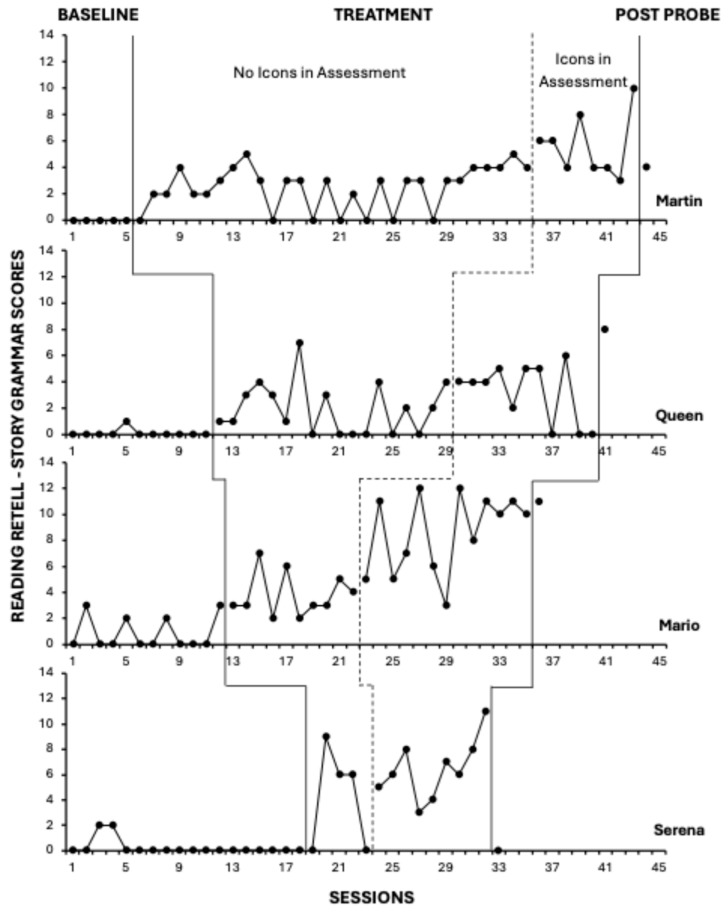
Reading comprehension results.

## Data Availability

The data related to this study are publicly available at LDbase.org. http://ldbase.org/datasets/25302270-dd4f-4a43-be71-2541fcd400cb (accessed on 22 May 2022).

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
