# Peer review of "Effect of Narrative Intervention with Strategy Instruction on the Listening and Reading Comprehension of Children with Autism"

_behavsci, 2025, doi:10.3390/bs15081020_

Round 1
Reviewer 1 Report
Comments and Suggestions for Authors
I appreciate the opportunity to review this interesting manuscript. This is a methodologically sound study that makes a valuable contribution to understanding narrative interventions for children with autism. The use of story grammar icons as a strategy shows promise. The authors appropriately reviewed the literature, justified for methods and reported findings.
Introduction/literature review
- The authors provide a solid research foundation linking narrative abilities to reading comprehension and highlight the specific challenges faced by children with autism in this domain. I would appreciate seeing an elaboration of the mechanism or theoretical explanation behind how narrative intervention could improve children with autism’s comprehension outcomes.
- Likewise, what are the distinctive mechanisms for listening and reading comprehension respectively? Studies have shown that for neural-typical people, different modalities might work better with different content types (Kürschner et al., 2006). For example, with less complex information, retention by listening and reading is similar, but with increased complexity, listening retention might be harder. What would you say about the level of complexity of your stories for administering to your participants?
[Kurschner, C. (2006). Construction of visio-spatial representations during listening and reading comprehension. ZEITSCHRIFT FUR PSYCHOLOGIE, 214(3),]
Methods
- The research design is appropriate for the research question. The experiment was a complex one but the authors did a good job in explaining and justifying the procedures with good clarity and conciseness. I have one question, though—what might be the potential impact on experimental control given the modification to add icons during assessment?
Results
- I appreciate the clear reporting of analyses and excellent visualization of results. Can you elaborate on how individuals responded to the intervention differently?
- I am alarmed to see the charts of listening and reading for Martin and Mario were nearly identical. Are there issues with the charts? If not, I hope to see an explanation of how these were possible.
Discussion
- The discussion is quite thorough, but in response to my comment #2, there needs to be an even more thorough discussion on why reading comprehension showed better maintenance than listening comprehension, with comparisons of research on autism and neural-typical populations.
- Similarly, in response to my comment #4, please discuss how participant characteristics influenced outcomes that might lead to clinical implications.
Other minor issues:
- Line 569: "For them, the intervention faded the prompts to quickly" - Should be "too quickly"
- Line 142: "24 different stories exemplars" - Should be either "24 different story exemplars" or "24 different stories as exemplars"?
- Line 584-585: "...when they need it, how they need it, and for as long as they need." - Missing "it"?
- Line 632: Is "Pro 36154" a complete/correct reference number? Kindly check.
- References : Inconsistent referencing style and formatting of DOIs. For example, some use "doi:" prefix while others use "https://doi.org/" format.
I would hope to see this paper published after a moderate level of revision.
Author Response
RESPONSES TO REVIEWER 1
Introduction/literature review
- The authors provide a solid research foundation linking narrative abilities to reading comprehension and highlight the specific challenges faced by children with autism in this domain. I would appreciate seeing an elaboration of the mechanism or theoretical explanation behind how narrative intervention could improve children with autism’s comprehension outcomes.
Response: We added a paragraph of theoretical and empirical explanations for narrative interventions for children with autism, beginning at line 109.
- Likewise, what are the distinctive mechanisms for listening and reading comprehension respectively? Studies have shown that for neural-typical people, different modalities might work better with different content types (Kürschner et al., 2006). For example, with less complex information, retention by listening and reading is similar, but with increased complexity, listening retention might be harder. What would you say about the level of complexity of your stories for administering to your participants?
[Kurschner, C. (2006). Construction of visio-spatial representations during listening and reading comprehension. ZEITSCHRIFT FUR PSYCHOLOGIE, 214(3),]
Response: Unfortunately, we were unable to get a copy of the suggested article. Nonetheless, we spent a great deal of time building out some of the relevant mechanisms. For this study, and our work in general, we use listening and reading passages that have equivalent complexity, to isolate the impact of modality. We have outlined this in the introduction (lines 55-88).
Methods
- The research design is appropriate for the research question. The experiment was a complex one but the authors did a good job in explaining and justifying the procedures with good clarity and conciseness. I have one question, though—what might be the potential impact on experimental control given the modification to add icons during assessment?
Response: The answer to this question is now in multiple places in the discussion section, but particularly well described in the Limitations (lines 796-811).
Results
- I appreciate the clear reporting of analyses and excellent visualization of results. Can you elaborate on how individuals responded to the intervention differently?
Response: Information added to contrast baseline performance and intervention effects for each of the four participants in the Results section (pp. 11-14).
- I am alarmed to see the charts of listening and reading for Martin and Mario were nearly identical. Are there issues with the charts? If not, I hope to see an explanation of how these were possible.
Response: I cannot thank you enough for catching this error. We honestly could not recall what happened, but luckily it took us back to the raw data, which ultimately led to totally updated figures. We are so happy that you caught this and prevented a serious mistake. New figures pp. 11 and 13.
Discussion
- The discussion is quite thorough, but in response to my comment #2, there needs to be an even more thorough discussion on why reading comprehension showed better maintenance than listening comprehension, with comparisons of research on autism and neural-typical populations.
Response: There is a new paragraph in the Discussion about this, lines 660-669.
- Similarly, in response to my comment #4, please discuss how participant characteristics influenced outcomes that might lead to clinical implications.
Response: We added specific information about participants’ characteristics and baseline levels of story retell performance to the Discussion section (lines 640-650). However, because children with autism are highly variable and there were only four of them, a deeper analysis of associations between characteristics and outcomes is not possible with these data.
Other minor issues:
- Line 569: "For them, the intervention faded the prompts to quickly" - Should be "too quickly”
Response: Revised ‘to’ to ‘too’.
- Line 142: "24 different stories exemplars" - Should be either "24 different story exemplars" or "24 different stories as exemplars"?
Response: We revised the sentence to “This fading procedure, coupled with Story Champs’ 24 different stories, make it potentially better…”.
- Line 584-585: "...when they need it, how they need it, and for as long as they need." - Missing "it"?
Response: Revised sentence to: “Children can potentially use the icons as a just-in-time support when they need it, how they need it, and for as long as they need it.”
- Line 632: Is "Pro 36154" a complete/correct reference number? Kindly check.
Response: This is the number assigned to the IRB application by the Institutional Review Board of University of South Florida. No changes are necessary. (p. 19)
- References : Inconsistent referencing style and formatting of DOIs. For example, some use "doi:" prefix while others use "https://doi.org/" format.
Response: We have revised all references to include the website address for the DOIs. However, we acknowledge they are not yet perfect. It is after 1am for me and I can't do any more fixes on references tonight. I know we will have another opportunity to polish them before publication.
Reviewer 2 Report
Comments and Suggestions for Authors
Line 182: It would be helpful to briefly describe the tools used—especially NLM, which appears to be a key component of the study. While it is later linked to the CUBED assessment suite, an earlier and clearer explanation would enhance reader comprehension. The current phrasing suggests that NLM is widely known and routinely used, which may not necessarily be the case.
Line 187: Please briefly clarify what Level 3 of the VB-MAPP corresponds to in terms of developmental or communicative milestones.
Lines 189–190: It would be useful to specify what the parent and clinician reports are based on: for example, medical history, parental questionnaires, adaptive functioning scales, or other sources.
Line 276: Including an example of these illustrations in an appendix (referred to later as “icons,” if correctly understood) would be valuable for the reader.
Line 373: Please clarify whether this refers to the same narrative used earlier in the study or to a different one.
Line 427: It is assumed that the score reflects the child’s recall of story elements. However, it would be helpful to provide more detail regarding the scoring procedure, particularly how performance on macrostructure and microstructure recall was assessed. This aspect remains unclear in the current version of the article.
Author Response
RESPONSES TO REVIEWER 2 COMMENTS
Line 182: It would be helpful to briefly describe the tools used—especially NLM, which appears to be a key component of the study. While it is later linked to the CUBED assessment suite, an earlier and clearer explanation would enhance reader comprehension. The current phrasing suggests that NLM is widely known and routinely used, which may not necessarily be the case.
Response: We added more description of the CUBED to the introduction as it facilitates the theoretical links between LC and RC (lines 69-88) and included website where it can be downloaded for free (line 333).
Line 187: Please briefly clarify what Level 3 of the VB-MAPP corresponds to in terms of developmental or communicative milestones.
Response: Added information about level 3 of the VB-MAPP: “(i.e., receptive and expressive language and learning milestones representative of children aged 30–48 months; Padilla & Akers, 2021)”. (lines 243-244)
Lines 189–190: It would be useful to specify what the parent and clinician reports are based on: for example, medical history, parental questionnaires, adaptive functioning scales, or other sources.
Response: Revised to improve clarity and specificity: “Per parent and clinician verbal reports to the research team, none of the participants exhibited severe problem behavior...”. (line 246).
Line 276: Including an example of these illustrations in an appendix (referred to later as “icons,” if correctly understood) would be valuable for the reader.
Response: All the visual material used in the Story Champs intervention are copyrighted and should not be included in research publications; however, examples of these are easily found online or YouTube. In case it is useful, we have added the website where the visuals can be seen to the manuscript (lines 327-328).
Line 373: Please clarify whether this refers to the same narrative used earlier in the study or to a different one.
Response: We clarified that the post-probe used a novel story. It now reads, “At the end of the intervention condition, using a novel story, we conducted a post-probe without the story grammar icons available to determine the extent to which generalization to the far-transfer context occurred.” (line 416).
Line 427: It is assumed that the score reflects the child’s recall of story elements. However, it would be helpful to provide more detail regarding the scoring procedure, particularly how performance on macrostructure and microstructure recall was assessed. This aspect remains unclear in the current version of the article.
Response: We added more specifics in the Method section around scoring, lines 363-378.
Reviewer 3 Report
Comments and Suggestions for Authors
First of all, I would like to thank you for the opportunity to review this interesting and valuable contribution. The aim of this study was to investigate the effect of the Story Champs intervention on the listening and reading retells of children with autism. The manuscript is well-structured, grounded in solid theoretical foundations, and presents a thoughtful and methodologically sound research approach. In my opinion, this is a timely and important topic worthy of exploration. After addressing a few additional comments, the manuscript will be suitable for publication.
- Due to the small number of participants, please indicate clearly in the title, as well as in the abstract and main text, that this is a pilot experimental study.
- Line 15: Please specify the age of the children more precisely. The term "school-aged children" is too broad given the small sample size.
- At the end of the introduction, following the research aims, please include hypotheses or research questions.
- Lines 180–183: Did the authors define any exclusion criteria? If so, please provide them.
- Did the authors obtain approval from an Ethics Committee for conducting the study? If yes, please provide the approval number.
- Which statistical software was used to analyze the data?
- The discussion section begins with an unnecessary repetition of the study's aim.
- Out of the 50 references, only 8 are from the last 5 years. There are no sources from 2025, 2024, or 2022. There is 1 reference from 2023, 3 from 2021, and 4 from 2020. The authors may consider including more recent studies, if available.
Author Response
RESPONSES TO REVIEWER 3 COMMENTS
- Due to the small number of participants, please indicate clearly in the title, as well as in the abstract and main text, that this is a pilot experimental study.
Response: While I understand this recommendation, we believe it is a matter of preference, familiarity, and do not feel adding “pilot” is the right thing to do. The title is already very specific and long. The study is a strong experimental design that is perfectly suitable for causal inference (i.e., controlling for threats to internal validity). The designation of “pilot” is unrelated to the sample size, despite some fields understanding it that way. Pilot is best given to studies that lack the rigor to minimize threats to internal validity (e.g., single group studies, convenience sample comparison groups). A multiple baseline across four participants is often misunderstood as a “quasi-“ experimental design because participants are not randomly assigned to treatment and control groups; however, random assignment is the strongest method for reducing threats to internal validity for group studies, but not for single case studies. As explained under Research Design, single case design’s quality indicators are about the number of data points and number of opportunities to observe level, trend, and variability changes across time; thereby providing readers and understanding of how they should evaluate this design. In this case, the minimum for a strong study is three opportunities for effect replication and our study includes four, exceeding that minimum standard. Although we know SCED is not well understood in every country, but it should be noted that SCED orginations in the Science of Behavior and Experimental Analysis of Behavior. Given the journal title and considering this study, it is inappropriate to label it as a “pilot”. Furthermore, both authors are veteran research methodologists and we teach advanced research methods (both group and single case designs, among others) to doctoral students. We have also co-authored seminal works in this research methodology.
- Line 15: Please specify the age of the children more precisely. The term "school-aged children" is too broad given the small sample size.
- Response: We revised the Abstract to improve specificity of participant ages. It now reads, “Four children with autism aged 7- and 9-years-old participated in this multiple baseline across participants single case experimental design study.” (lines 16-17).
- At the end of the introduction, following the research aims, please include hypotheses or research questions.
Response: We have converted the purpose statement to research questions (lines 203-207).
- Lines 180–183: Did the authors define any exclusion criteria? If so, please provide them.
Response: We added exclusion criteria information: “Children who engaged in behavior(s) that routinely interrupted intervention delivery and threatened the health and/or safety of themselves or others (e.g. physical aggression), as well children who lived outside of the U.S., were excluded from the study.” (lines 236-237.)
- Did the authors obtain approval from an Ethics Committee for conducting the study? If yes, please provide the approval number.
Response: The IRB approval number/reference is provided in the Institutional Review Board Statement section at the end of the manuscript (lines 828-829).
- Which statistical software was used to analyze the data?
Response: The single-case experimental methodology required structured visual analysis of the observed data and did not involve use of statistical software. We have added statistical analyses and software is described in lines 471-472.
- The discussion section begins with an unnecessary repetition of the study's aim.
Response: We reduced the study aim content, reminded the reader of the research question and justification, then got to the results quicker (lines619-624).
- Out of the 50 references, only 8 are from the last 5 years. There are no sources from 2025, 2024, or 2022. There is 1 reference from 2023, 3 from 2021, and 4 from 2020. The authors may consider including more recent studies, if available.
Response: To the extent possible, we revised the paper to include more contemporary research. Changes have been made to in-text citations and the Reference list. However, many of the critical arguments are related to seminal works from the 70s, 80s, and 90s. This is not an area that receives a ton of attention as of late.
Round 2
Reviewer 1 Report
Comments and Suggestions for Authors
I thank the authors for the great effort in revising the manuscript based on reviewers' comments. It is a much improved version with greatly enhanced clarity. I am happy with all the responses and changes.